# Harnessing the Power of Integrated Behavioral Health to Enhance Insomnia Intervention in Primary Care

**DOI:** 10.3390/jcm13185629

**Published:** 2024-09-23

**Authors:** Rebecca L. Campbell, Ana J. Bridges

**Affiliations:** 1Department of Psychology, University of Arizona, Tucson, AZ 85721, USA; 2Department of Psychological Science, University of Arkansas, Fayetteville, AR 72701, USA; abridges@uark.edu

**Keywords:** sleep medicine, treatment barriers, prevention, integrated care, behavioral health

## Abstract

Insomnia is prevalent in primary care and associated with co-morbid physical and mental health conditions and poor health outcomes. While there are effective treatments for insomnia in specialty mental health care, many patients have difficulty accessing these interventions. To begin, patients do not always report their sleep challenges to physicians; meanwhile, primary care providers often do not screen for insomnia symptoms. Furthermore, patients may experience several barriers to accessing specialty care for insomnia treatment, such as a limited number of available providers, financial burden, lack of transportation, and low perceptions of treatment effectiveness. Primary care behavioral health (PCBH) is well-equipped to address the challenges of accessing evidence-based care for insomnia through (1) identifying sleep issues, (2) providing psychoeducation on the possible treatments for insomnia, (3) intervening with poor sleep habits and acute insomnia early to prevent chronic insomnia, and (4) delivering appropriate evidence-based interventions for chronic insomnia. Primary care clinics should leverage behavioral health providers to increase screening and embed interventions into routine care for the benefit of improved outcomes for patients with insomnia and other sleep challenges.

## 1. Introduction

An estimated 10–15% of Americans have chronic insomnia, although only 1.2% have been formally diagnosed [1]. The DSM-5R defines insomnia as the difficulty falling asleep, staying asleep, or waking up too early despite adequate sleep opportunity [2]. Insomnia is commonly found in primary care settings, with a third of patients meeting criteria for insomnia, about half reporting daytime sleepiness, half experiencing occasional insomnia, and 19% suffering from chronic insomnia [3,4]. Furthermore, prevalence rates of insomnia appear to be increasing [5]. Insomnia is associated with substantial medical costs [6,7]. As an added challenge, insomnia is often observed with various comorbidities such as hypertension, pain, depression, alcohol consumption, smoking, stress, heart failure, coronary heart disease, gastroesophageal reflux disease, irritable bowel syndrome, diabetes, and anxiety [4,8,9,10,11,12]. One study found people with sleep disturbances have a mortality risk three times higher than those without, and a higher rate of suicidality [8].

While, historically, specialty mental health care clinics have been the place for patients to receive behavioral treatments for insomnia, the full demand for treatment is not being met. Fortunately, primary care behavioral health (PCBH) may be an excellent way to address gaps in care. With its unique vantage point, primary care can effectively identify and treat patients needing assistance and situate their sleep issues within the comprehensive context of comorbidities and functional challenges.

## 2. The Power of Primary Care Behavioral Health

PCBH is well-suited to meet the needs of patients and providers alike. The PCBH model involves a mental health professional on the primary care team who works to manage behavioral, mental, and physical health conditions using a biopsychosocial approach [13]. The mental health professional, referred to as a behavioral health consultant (BHC), works side-by-side with primary care providers to improve services throughout the primary care clinic. BHCs take a generalist approach to their work, seeing a wide array of patients, and ensure behavioral health services are an accessible part of routine care [14]. For this reason, BHCs are well-situated to address insomnia in primary care clinics. However, the presence of a BHC alone is not sufficient to have an impact on screening or interventions for sleep issues [15]. To improve sleep outcomes, it is recommended that BHCs focus on four main goals: (1) identifying sleep issues, (2) providing psychoeducation on the possible treatments for insomnia, (3) intervening with poor sleep habits and acute insomnia early to prevent chronic insomnia, and (4) delivering appropriate evidence-based interventions for chronic insomnia (Figure 1). We review each in turn.

The diagram illustrates the core elements of an integrated behavioral health approach to addressing insomnia. The four interrelated aims include (1) identifying sleep issues, (2) providing psychoeducation on the possible treatments for insomnia, (3) intervening with poor sleep habits and acute insomnia early to prevent chronic insomnia, and (4) delivering appropriate evidence-based interventions for chronic insomnia.

### 2.1. Identifying Sleep Issues 

It is unlikely that many patients will independently seek help for sleep issues. One study found that only about 40% of people suffering from insomnia sought help from a healthcare professional [16]. Patients may believe insomnia is a trivial problem they should be able to cope with alone [17]. In fact, adults with poor sleep tend to have more rigid beliefs about the controllability and the consequences of poor sleep than adults with good sleep [18], suggesting they are unlikely to seek help for something they believe to be in their control. Most patients have a limited knowledge of possible treatment methods or do not consider the methods they are aware of to be effective or appealing [17]. Since patients are less likely to reach out for support, primary care providers need to broach the subject of sleep challenges with their patients.

Despite the prevalence of sleep disorders in primary care, screening for such challenges is not routine [19]. In one study, 366 primary care patients were interviewed about insomnia symptoms in the waiting room before their appointment. About 75% of patients met criteria yet not a single patient reported or was screened for sleep challenges [20]. This may be because providers do not consider sleep challenges to be a high priority concern, do not have enough time, or assume patients will mention any challenges they might be experiencing [15,21], thus placing the burden of reporting on patients. Furthermore, many providers have the impression insomnia is typically a symptom of something else, rather than a disorder unto itself, and thus, its treatment is deprioritized [21].

Screening and diagnostic accuracy of sleep disorders is imperative, especially because many sleep disorders share symptoms, such as experiencing daytime sleepiness. Disorder-specific interventions for sleep problems are not interchangeable. A misdiagnosis may result in ineffective treatment and frustration on the part of the patient and the provider, reinforcing the belief that sleep treatments are not effective and making it even less likely a patient will seek appropriate help for sleep challenges. Fortunately, several brief evidence-based screening approaches are available (Table 1). In addition, Goodie and Hunter [22] proposed a brief screening focusing on the history of sleep problems, pre-sleep behaviors, the sleep environment, the impact of the sleep disturbance on the person’s functioning, and potential diagnostic rule-outs. This allows for the diagnostic clarity necessary to make informed intervention recommendations. For more complicated cases, a structured clinical interview such as the Structured Clinical Interview for Sleep Disorders—Revised (SCISD-R) [23] may clarify diagnostic uncertainty.

Time is a valuable and scarce resource in the primary care setting, so adding patient insomnia screenings to the duties and responsibilities of primary care providers is inadvisable. Instead, this critical task could be delegated to a BHC or a BHC assistant, i.e., to someone with limited, yet targeted, training in behavioral sleep medicine. For instance, BHCs or BHC assistants can administer a brief insomnia screener such as the Insomnia Severity Index (ISI) [24]. Patients who meet a specified cutoff could be referred for a same-day appointment with the BHC for further evaluation via more detailed self-report measures or a structured clinical interview. If additional diagnostic clarity is needed (e.g., to rule out sleep apnea or narcolepsy), the BHC could facilitate a referral to a specialty clinic for a sleep study. Otherwise, or in parallel, the patient and BHC could work together on behavioral interventions for identified problems.

### 2.2. Providing Psychoeducation

If a patient decides to consult with a professional about their sleep challenges, they are most likely to relay this information to their primary care provider. Research has shown that, of those patients who eventually consulted a medical professional about their insomnia, 82.7% consulted their general practitioner [24]. However, many practitioners are not confident in their knowledge of sleep and sleep disorders [15,30,31]. As a result, they often default to pharmaceutical interventions because they believe that is what their patients are expecting [31,32]. If behavioral strategies are offered, they are typically sleep hygiene recommendations, though practitioners perceive sleep hygiene as ineffective [32], and for good reason. Sleep hygiene is not recommended as a stand-alone treatment for chronic insomnia [33,34]. 

Many adults understand sleep to be important for their health [35] but may have conflicting priorities or misunderstandings about what can be done about their sleep. Patients and providers alike have a limited understanding of the possible treatments for insomnia, either not seeing them as being effective, perceiving them as too resource intensive, or not knowing about these treatments at all, instead opting to “deal with it” on their own [15,17,21,32]. Psychoeducation and possible referrals from a BHC about sleep disturbances may highlight paths and options not otherwise considered. Additionally, a BHC could provide in-clinic training opportunities for medical providers and clinic staff on insomnia and treatment options, allowing the team to better serve their patients. While patient awareness of treatment options does not equate to immediate change, it does provide directions when a patient is ready for treatment. 

### 2.3. Intervening Early

Primary care providers agree prevention of sleep disorders is important; however, preventative efforts are often deprioritized over other health concerns such as diet or exercise [30,31], suggesting there is room for improvement. Here, the BHC is an invaluable asset. Chronic insomnia is, by definition, a behavioral disorder. According to the 3P model of insomnia [36], patients have predisposing factors that increase the likelihood of insomnia (such as genetics) and precipitating factors (such as life stress or changes) that result in difficulty sleeping. However, the maintaining elements, and thus the targets of behavioral treatment, are perpetuating behaviors such as trying to compensate for a short night of sleep with naps, going to bed early, or sleeping in, leading to increased insomnia symptoms. These behaviors are easily identifiable and modifiable when working with a BHC. For instance, when patients are experiencing a stressful life event, a period in which sleep disturbances are to be expected, brief psychoeducation about the negative effects of compensatory sleep behaviors may be enough to prevent the development of chronic insomnia. Furthermore, modifying sleep-related behaviors outside of stressful life events may also prove beneficial. For instance, adjusting poor sleep hygiene, such as drinking caffeine in the afternoon, may curb the risk of developing insomnia (see Table 2 for details of sleep hygiene).

Another avenue to prevent chronic symptoms may involve a BHC working with the primary care provider to manage and monitor medications that may have impacts on sleep, such as psychiatric medications [37]. For instance, BHCs can work with patients to monitor side effects, especially those related to sleep, and then work with the provider to determine if there should be a dosage change, medications should be taken at a certain time of day, or the patient needs to be switched to a different medication.

### 2.4. Delivering Evidence-Based Treatment

Cognitive Behavioral Therapy for insomnia (CBTi) is the first-line treatment recommended by the American College of Physicians [38]. This treatment has been found to be as effective as medications in the short-term and more effective in the long-term [39]. Despite the outstanding effectiveness of CBTi, the patients who would benefit most from this intervention often never make it to specialty care where it is commonly delivered [38,40]. Many primary care providers have limited knowledge of the availability of specialty care treatments and thus are unlikely to refer [32]. Even when referrals to specialty care are made, patients often fail to follow through [41]. Treatment of insomnia is most often provided by a psychologist with specific training in behavioral sleep medicine and/or cognitive behavioral therapy [39]. CBTi typically spans 6–8 sessions and includes psychoeducation on the development and maintenance of insomnia, sleep restriction, stimulus control, and cognitive techniques to control anxious thoughts about sleep [39]. Relaxation training is also a common component of CBTi (see Table 3 for a brief explanation of the components of CBTi). 

The challenges of accessing evidence-based insomnia care continue even after a patient is properly identified and referred. Firstly, there is a shortage of trained CBTi providers, resulting in long wait times or extensive travel requirements to be seen [21]. When a patient is seen, they must attend 6–8 h long weekly sessions—a hefty time commitment when considering factors such as work schedules and childcare. These sessions occur within operating hours of the specialist, which may make it difficult or impossible for a patient to fit it into their schedules. The financial burden posed by two months of mental health treatment can also be considerable, resulting in a major hurdle for low income or uninsured patients. 

Even when a patient successfully makes it into specialty care, completing treatment can be difficult. Patients report numerous external pressures, such as work schedules or childcare, that make it difficult to modify sleep routines, habits, and schedules [21]. These difficulties may result in premature termination of treatment or reduced treatment doses. Patients are more willing to adhere to an “easy” treatment that fits in with their busy lifestyles [21]. One could argue that CBTi is the best treatment option, and therefore, patient motivation and access should be the targets of change. On the other hand, a treatment can only be effective when it reaches the target population. Currently, there are significant barriers leaving many insomnia sufferers without the support they need. Patients who cannot access specialty mental health treatment for insomnia are nevertheless entitled to evidence-based treatment and support. 

There may be instances in which interventions with the BHC are warranted and necessary, such as subclinical or acute cases of insomnia or in cases where specialty mental health referrals are not an option. Here, a stepped-care approach may be most effective [42]. Murawski and colleagues conducted a systematic review and found interventions that employed strategies such as stimulus control, meditation, breathing, and stress management improved sleep quality in adults with poor sleep who did not meet criteria for a sleep disorder [43]. Studies have shown abbreviated [44] and group [45] CBTi are effective in a primary care setting. In a systematic review, Cheung and colleagues found small to moderate effects on insomnia severity, sleep quality, sleep onset latency, and wake after sleep onset when stepped-care CBTi was administered in primary care settings [46]. Abbreviated CBTi, protocols that include at least four sessions of stimulus control and sleep restriction appear to have the strongest effects [47].

CBTi, administered in its entirety or specific components, is an effective treatment for insomnia [40]. One option suited to the primary care setting may be a CBTi group conducted by the BHC, especially if insomnia is a common concern among patients of the clinic. CBTi groups have been shown to be effective [21] and allow for multiple patients to receive services at the same time, thereby increasing the reach of services. These groups can be structured as standard CBTi, in which members join at the beginning and progress through the 6–8 sessions with new groups starting after the completion of the last. Alternatively, there can be a more flexible drop-in format in which groups are open to all patients and there are different themes for each group session, allowing patients to choose what is most relevant to them.

The specific components of CBTi have been shown to be effective in treating insomnia as well [40]. Therefore, using components such as sleep restriction, stimulus control, and relaxation in a primary care setting may be useful treatment approaches [22]. Use of components should be informed by the mechanisms of each component and how they relate to the case conceptualization. Strategies could be informed by the patient’s current behavior patterns. For instance, a patient who engages in a lot of activities in bed may see the largest benefit from stimulus control, while a patient who experiences physiological arousal before bed may see stronger effects from relaxation training (see Table 3).

There is some evidence that a single session of CBTi may be effective in treating acute insomnia [48]. In this hour-long appointment, patients receive psychoeducation about insomnia and learn about sleep restriction and stimulus control. Patients are then given sleep diaries and instructed on how to calculate sleep efficiency, so they can independently adjust their sleep schedule. Participants who received the single session of CBTi had significantly lower ISI scores compared to a waitlist control at a one-month follow-up, and 60% had remitted compared to only 15% of the control group. Replication of this study in primary care is much needed.

There are other treatments beyond CBTi that have demonstrated effectiveness in sleep disorders. For instance, cognitive refocusing therapy (CRT) is a technique that targets anxious thoughts before falling asleep [49]. In a single session, the BHC and patient brainstorm a mental activity that is engaging enough to maintain attention, but not emotionally or physically arousing, such as reciting the lyrics to a favorite song. In Gellis et al., undergraduates were given a single session of CRT and sleep hygiene. They found participants in the CRT+ sleep hygiene group improved more than the sleep hygiene only group, when controlling for anxiety and non-sleep related depression [49]. This finding replicated a four-session trial with a veteran sample [50]. Importantly, the undergraduates reported the treatment was reasonable and they would recommend it to others [49]. CRT may serve best as a “just-in-time” intervention for patients who have started noticing difficulty falling asleep but who have not yet developed an entrenched pattern of compensatory behaviors. Furthermore, this approach targets anxious thoughts in general, not just anxious thoughts about sleep, so CRT may be ideal for anxious individuals, particularly in times of stress, who begin struggling to fall asleep due to racing thoughts. Future work is needed to demonstrate the effectiveness of CRT in a primary care setting, distinguish if it is useful for chronic or acute insomnia, and determine if it works better for anxious or non-anxious sleepers.

Anxious individuals with sleep disturbances may also benefit from constructive worry [51,52]. This treatment entails writing down worries and problem-solving for those worries in the early evening. One study compared a behavioral intervention (stimulus control and sleep restriction) group to a behavioral and constructive worry group, finding both resulted in fewer insomnia symptoms and less worry [52]. Constructive worry and imagery distraction have been shown to reduce worry and improve sleep outcomes [51].

## 3. Clinical Vignette

M is a 33-year-old White woman who reported to her primary care provider for complaints of gastrointestinal upset. In the waiting room, she completes the ISI. Her score of 18 suggests she is experiencing moderate clinical insomnia symptoms. Her primary care provider asks if she would be open to meeting with the BHC to discuss her sleep challenges. M agrees and her provider initiates a warm handoff to the BHC.

### 3.1. Identifying 

The BHC starts by reviewing M’s responses on the ISI and learns that M has been taking between 45–120 min to fall asleep and is awake for 30–60 min after sleep onset at least four times a week. M reports coping with the sleep loss by sleeping in on nights where she does not get enough sleep and taking long naps when possible. She also noted that she works from home and will often engage in activities in bed such as working, eating, and reading. M noted sleep issues began six months ago when her partner ended their relationship and moved out of M’s apartment. M describes a flexible work schedule that allows her to go to bed late (2400–0100) and wake up late (0900–1100) resulting in a 9–10-h sleep opportunity, although M only gets about 6 h of sleep on a typical night. M denied symptoms associated with sleep apnea such as snoring and gasping at night.

The BHC conceptualizes that M’s breakup was likely a precipitating factor in the development of insomnia and symptoms are perpetuated by the extended sleep opportunity, daytime napping, and activities in bed.

### 3.2. Psychoeducation 

Firstly, the BHC validates the challenges that M is experiencing and explains the relation between insomnia and stress. She asks M if this conceptualization is in line with her experience. When M agrees, the BHC asks if she can share some information relevant to M’s sleep. The BHC describes stimulus control, specifically how engaging in activities such as working, reading, worrying and lying awake trying to fall asleep weakens the association between the bed and sleep. Finally, the BHC describes how the sleep drive and circadian rhythms work together to regulate sleep and wake. She explains how napping during the day can deplete the sleep drive and make it difficult to sleep at night.

She asks M if she has any questions or concerns. When M states she understands, the BHC then asks her if she thinks any of her behaviors are contributing to her insomnia. M identifies spending time in bed while awake, sleeping in, and napping. The BHC agrees with this assessment and begins planning next steps with M.

### 3.3. Treatment

When building the plan for M, the BHC prioritizes stimulus control and a consistent sleep schedule. M agrees to only use her bed for sleep and sex and to only get into bed when she is feeling sleepy. Together they determine a 0930 waketime will work best for M’s circadian tendency and lifestyle. M also agrees to stop napping. Over the course of the next few weeks, the BHC regularly checks in with M via a secure portal. Check-ins include administration of the ISI, confirming adherence to the initial behavior plan, and troubleshooting difficulties.

At her six-week telehealth follow-up, M’s ISI score is 10, suggesting she is still experiencing mild insomnia symptoms. She reported gains in her sleep have “leveled off” such that she is sleeping through the night, but it still takes her at least 30 min to fall asleep. Based on a chart review, the BHC notices M has a history of generalized anxiety disorder and hypothesizes that anxious and racing thoughts are the main contributor to her difficulty falling asleep. She discusses her hypothesis with M and asks questions related to M’s experience falling asleep. Using a shared-decision approach, they agree to focus on managing anxiety by implementing constructive worry. The BHC continues to monitor sleep and adds an additional measure of anxiety to check-ins. After two more weeks, M reports being able to fall asleep within 30 min most nights. The BHC instructs M to follow up with her or with M’s primary care provider if symptoms return. The BHC also sets a reminder in M’s electronic medical record to check in with her via the secure portal in a few months.

## 4. Discussion

BHCs practicing in PCBH settings can help increase access to behavioral sleep treatments by identifying sleep issues, providing psychoeducation, intervening early when sleep problems first emerge, and adapting evidence-based treatments to be delivered in primary care. Doing so will result in a significant reduction of the burden of insomnia on primary care patients.

### 4.1. Site-Specific Considerations

It is important for clinics to adapt the application of these four components to fit their specific needs. Adaptions can vary depending on the population served, clinic resources, and clinic structure. While a full review of possible adaptations is beyond the scope of this paper, we discuss the age of the target population as an example.

Sleep generally, and insomnia specifically, looks different across the lifespan. Older adults experience less deep sleep and more wake after sleep onset and light sleep as they age [53]. They may also report more sleep disturbances related to medication use or comorbid medical concerns [53]. Children, on the other hand, need substantially more sleep, including daytime naps [54]. Sleep disorders such as insomnia are not uncommon in children. However, they may be underdiagnosed in primary care, possibly because of minimal screening and lack of caregiver awareness [55].

While behavioral interventions to treat insomnia are still recommended over the use of medications, there are different treatment modifications to consider [56]. For instance, older adults have a high risk for falls, thus the stimulus control recommendations to get out of bed if awake may not be safe. Instead, a patient might consider modifications such as countercontrol [57]. Rather than getting out of bed, patients create a clear distinction between their posture and environment when in bed to sleep versus when they are awake in bed. 

When working with children, screening procedures may vary. It is recommended to have the patient and their caregiver complete a sleep diary to facilitate discussions about sleep [56]. It is also important to inquire about caregiver support in sleep-related behaviors [58,59]. When treating insomnia in children and adolescents, behavioral interventions are effective [60,61,62,63]. Notably, sleep restriction is typically implemented only after other behavioral strategies have been exhausted [60]. 

### 4.2. Facilitators and Barriers

Getting started with targeted insomnia assessment and treatment is a challenging undertaking for any clinic. When initiating similar programs, common barriers include limited awareness and knowledge on the part of patients and providers, limited resources and time, an increase in workload, provider beliefs, and low interest, motivation, and prioritization [64,65,66,67]. At the outset, there may be large time and financial commitments as key personnel are trained in the four components and clinics experiment in how best to introduce this into the current workflow. However, implementation is facilitated by several factors such as educational opportunities for patients and providers, good collaboration and communication, good record systems, individualized treatments, and the recruitment of specialized workers [64,65,66,67].

The impact of these facilitators and barriers will vary by clinic. The persons championing insomnia treatment may need to dedicate time to developing a clinic atmosphere that is receptive to changes to improve insomnia treatment. This may look like organizing training opportunities about the impacts of insomnia, best clinical practices for intervention, and referral pathways. Because a system-wide change can be costly in the front end, it could be important to get a snapshot of how many patients at a given clinic might benefit from these services to determine how large of an impact targeting insomnia might have. Increasing patient awareness about insomnia (e.g., through passive psychoeducational handouts distributed in waiting rooms) might also be a strong beginning step.

Once the clinic is on board, the next step might be to identify (1) who is involved, (2) in which component they are involved, and (3) how personnel will communicate with one another. Considerations here may include time pressure, expertise, and at what point they encounter a patient during a visit. For instance, administration of a screener measure can take place at several timepoints throughout a patient visit. It could be included in the initial appointment paperwork or handed out by front desk staff. Alternatively, it may be more feasible to have a medical assistant or even a physician administer the measure for a quick review and warm handoff to the BHC. Initial attempts should be iterative, take note of bottlenecks, overburdened personnel, or communication lapses, and use provider and patient feedback to make changes.

### 4.3. Research Needed

There are treatments that may be appropriate for a PCBH setting. However, these interventions need to be refined and tested in primary care. Implementation of evidence-based sleep treatments in primary care may need to be modified to consider both the strengths and limitations of the primary care setting. Primary care requires treatments of shorter duration and intensity but can also leverage the expertise of an interdisciplinary care team and coordinated case management. The scholarly research would benefit from dismantling studies of sleep treatments, such as CBTi, to determine what components have the largest impacts and for whom. In addition, studies can examine whether BHCs or other primary care extenders may be best suited to administer these treatment components. Future work should also examine the role of comorbid disorders as moderators of treatment outcomes. For instance, a cognitive sleep intervention may be more impactful for a patient who experiences a lot of physiological arousal or rumination, while a less anxious person may find behavioral interventions such as sleep restriction more useful.

Lastly, this review is designed to integrate the evidence for PCBH and insomnia treatment and present it in such a way that it can be implemented in the real world. We suggested the components based on the existing evidence, but future research is needed. It is important to determine if the four components are weighted equally in terms of impact and burden. Such an investigation will guide clinics to focus resources on the component with the largest benefit. Additionally, it would be important to understand how components may interact with one another. While there is good work that supports each individual component, to our knowledge there is no work that examines these components (or similar ones) as a whole.

## 5. Conclusions

To meet the needs of primary care patients, further training for BHCs is needed. BHCs will need basic training in sleep assessment to distinguish between various sleep disorders or refer to needed screening or specialty care. Furthermore, training in interventions will also be necessary. After training BHCs, primary care clinics would benefit from implementing universal or indicated screening procedures to assess their clinic-specific needs. Screenings will need to occur with attention to logistics, placing as little burden on clinic staff as possible. All clinic staff should be aware of different treatments offered to facilitate referrals to the BHC.

PCBH is a promising approach to reduce the rate and impact of insomnia on a population scale. It is an opportunity to intervene early to reduce the likelihood of developing chronic issues and to show a path to recovery for those who are suffering. The accessible nature of PCBH is well-suited to provide interventions for a myriad of patients who may experience barriers to accessing specialty care. With proper training, BHCs are a welcome and much-needed addition to behavioral sleep medicine.

## Figures and Tables

**Figure 1 jcm-13-05629-f001:**
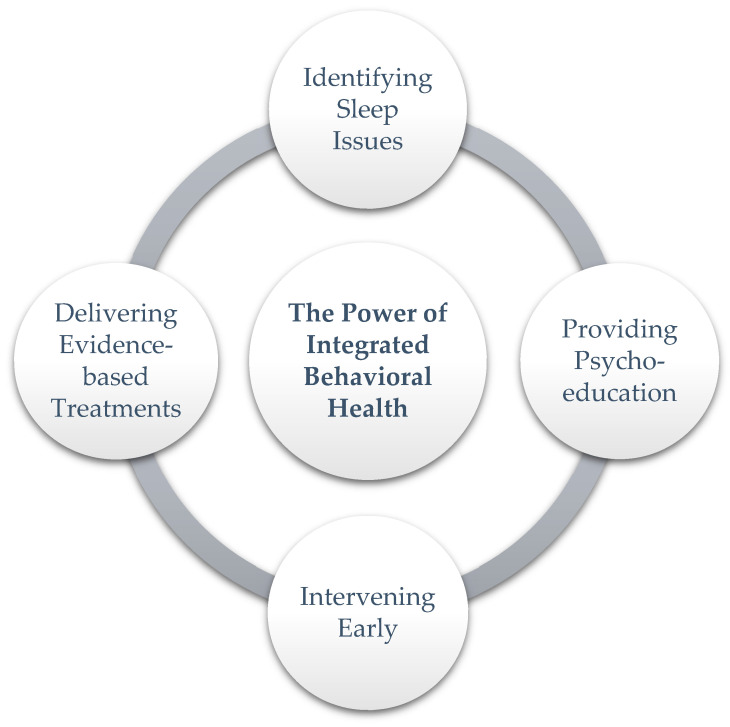
PCBH treatment targets.

**Table 1 jcm-13-05629-t001:** Screening measures.

Measure	Purpose	Description	Scoring
Insomnia Severity Index (ISI) [24]	Assess insomnia symptoms and related impairment in the past two weeks	7-item multiple-choice measure	Scores range from 0–28 with higher scores indicating more insomnia symptoms. Scores of 15 or greater indicate insomnia
Sleep Disorders Symptom Checklist (SDSCL) [25]	Screen for a variety of sleep disorders including insomnia, sleep apnea, phase delay/advance, restless leg syndrome, and parasomnias	25-item measure in which patients report the frequency of various sleep disorder symptoms on a scale of 0–4	Scores are divided into symptom-specific subgroups: obstructive sleep apnea, insomnia, narcolepsy, restless leg syndrome, and parasomnias. Higher scores are indicative of more frequent sleep disturbances
Pittsburgh Sleep Quality Index (PSQI) [26]	Assess global sleep quality in the past two weeks by taking into account sleep timing, medication use, daytime functioning, and environmental factors	19-item measure using a mix of multiple-choice items and open-response items. If available, 5 items are completed by a bed partner or roommate. Items are divided into 5 subscales	Scores range from 0–21 with higher scores indicating worse sleep quality. Scores of 5 or higher are considered poor sleep quality
Epworth Sleepiness Scale (ESS) [27]	Measure daytime sleepiness	8-item measure in which patients report how likely they are to fall asleep on a scale of 0–3	Scores range from 0–24 with ratings of 6 or above indicating high levels of daytime sleepiness
STOPBANG [28,29]	Determine risk for obstructive sleep apnea	8 yes/no items	Scores range from 0–8 with 5 or more endorsed symptoms indicating high risk for sleep apnea
Structured Clinical Interview for Sleep Disorders—Revised (SCISD-R) [23]	Distinguish between DSM 5-TR sleep disorders and collect information on medical history, mental health, medications and substances, and sleep schedule	2 sections related to medical history and sleep schedule and 8 disorder-specific sections that include questions, criteria, and presence ratings. Symptoms are rated “?” (insufficient information), 1 (absent), 2 (subthreshold), or 3 (threshold)	Each section uses skip logic based on DSM 5-TR diagnostic criteria to determine if patient meets criteria for each sleep disorder

This table summarizes six commonly used tools for assessing various aspects of sleep and sleep disorders. Each tool’s purpose, description, and scoring method are outlined. This list is non-exhaustive and should be used as a starting point when determining what measures work best for specific clinics.

**Table 2 jcm-13-05629-t002:** Sleep hygiene components.

Sleep Hygiene Component	Description	Utility
Wake activities	Exercise regularly Maintain a balanced diet and avoid eating too close to bed Reduce the intake of caffeine, nicotine, and alcohol, especially before bed	Good for addressing daytime activities that may make it harder to fall asleep or stay asleep.
Transition between wake and sleep	Create a bedtime space that is cool, dark, and quiet Build a bedtime routine to wind down before sleep Reduce watching the clock Reduce screen time before bed	Helpful in establishing a relaxed state in an environment conducive to sleep.

This table outlines essential components of sleep hygiene, focusing on wake activities and the transition between wake and sleep.

**Table 3 jcm-13-05629-t003:** Components of Cognitive Behavioral Therapy for Insomnia (CBTi).

CBTi Component	Description	Utility
Psychoeducation	Education about the nature of sleep (phases of sleep) 3P model of insomnia Medications (rebound insomnia)	Good to educate patients about what good sleep looks like, how insomnia develops, and how it is maintained. Provides a rationale for behavioral treatments. Provides information about the contraindications of pharmacotherapy-only approaches to treating sleep problems.
Sleep restriction	Restrict amount of time patient spends in bed to total sleep time (minimum 5 h window) Same bedtime and wake time, regardless of day of the week Build up need for sleep	Good for patients who try to “make up” for poor sleep by napping, sleeping a lot on weekends or days off, or going to bed early.
Stimulus control	Increase association of bed with sleeping Only use bed for sleep and sex Build good sleep-preparation habits Get out of bed when not asleep after 15–20 min	Good for patients who have a conditioned association between their bed and wakefulness.
Relaxation training	Progressive muscle relaxation Deep breathing Visualization Cognitive restructuring	Good for patients with anxiety or who become anxious at bedtime, anticipating a poor night’s sleep.

This table provides a brief overview of components of Cognitive Behavioral Therapy for Insomnia (CBTi), providing descriptions and the utility of each intervention.

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
