# Peer review of "Harnessing the Power of Integrated Behavioral Health to Enhance Insomnia Intervention in Primary Care"

_jcm, 2024, doi:10.3390/jcm13185629_

Round 1

Reviewer 1 Report

Comments and Suggestions for Authors

This review paper aims to discuss using a primary care behavioral health (PCBH) approach to address the insomnia issue. This is a good discussion which is similar to another important health issue, alcohol use disorders. I have the following suggestions for the authors to consider for the improvement of the paper. 

1. There are 4 components of PCBH mentioned in the paper: 1) identifying sleep issues, 2) providing psychoeducation on 16 the possible treatments for insomnia, 3) intervening with poor sleep habits and acute insomnia early 17 to prevent chronic insomnia, and 4) delivering appropriate evidence-based interventions for chronic 18 insomnia. 

Is this 4-component an evidence-based approach? Or there is evidence for individual component and the authors suggested combining the 4 components to treat insomnia using PCBH? 

If the 4-component is an evidence-based approach already, please give more references for it and discuss how it has been implemented in different countries/regions. 

If it is the latter case, the authors should point out this and have a brief discussion about future research studies that should investigate the efficacy of the 4-component approach in treating insomnia. 

2. The authors should discuss the cost or limitations related to implementing PCBH in primary care settings, especially financial costs, such as costs related to training PCBH professionals, the feasibility of organizing such service in the primary care settings, how it could be integrated into the service provided by physicians or nurses, etc.   

3. The discussion section should be strengthened. I suggest the authors use implementation science to discuss the facilitators and barriers to implementing PCHB, and as well as implementation strategies to enhance the facilitators and overcome the barriers. 

Look forward to reading the revised version of this paper.

Author Response

We sincerely appreciate the thoughtful feedback provided by the reviewers. Their insightful suggestions have greatly contributed to improving the clarity and quality of this manuscript. We are grateful for their time and effort in helping to strengthen our work. Below we address each set of comments. Edits are highlighted in the manuscript.

Reviewer 1

  1. There are 4 components of PCBH mentioned in the paper: 1) identifying sleep issues, 2) providing psychoeducation on the possible treatments for insomnia, 3) intervening with poor sleep habits and acute insomnia early to prevent chronic insomnia, and 4) delivering appropriate evidence-based interventions for chronic insomnia. 

Is this 4-component an evidence-based approach? Or there is evidence for individual component and the authors suggested combining the 4 components to treat insomnia using PCBH? 

If the 4-component is an evidence-based approach already, please give more references for it and discuss how it has been implemented in different countries/regions. 

If it is the latter case, the authors should point out this and have a brief discussion about future research studies that should investigate the efficacy of the 4-component approach in treating insomnia. 

Thank you for your insightful feedback. The proposed components are grounded in a robust body of research, with each element supported by individual studies. We have synthesized our understanding of the benefits of PCBH and insomnia treatment to connect existing insomnia research to the PCBH framework. For example, screening is a common practice for Behavioral Health Consultants (BHCs), and in the manuscript, we aim to provide specific examples of evidence-based measures to aid in this endeavor. To make this clear, we have added an additional paragraph in the discussion section explaining the limitation and suggesting future research.

“Lastly, this review is designed to integrate the evidence for PCBH and insomnia treatment and present it in such a way that it can be implemented in the real world. We suggested the components based on the existing evidence, but future research is needed. It is important to determine if the four components are weighted equally in terms of impact and burden. Such an investigation will guide clinics to focus resources on the component with the largest benefit. Additionally, it would be important to understand how each component may interact with one another. While there is good work that supports each individual component, to our knowledge there is no work that examines these components (or similar) as a whole.” (p. 11, line 397)

  1. The authors should discuss the cost or limitations related to implementing PCBH in primary care settings, especially financial costs, such as costs related to training PCBH professionals, the feasibility of organizing such service in the primary care settings, how it could be integrated into the service provided by physicians or nurses, etc.   

We concur that this addition significantly strengthens the manuscript. Accordingly, we have incorporated a dedicated section in the discussion to address this important issue.

“Getting started with targeted insomnia assessment and treatment is a challenging undertaking for any clinic. When initiating similar programs, common barriers include limited awareness and knowledge on the part of patients and providers, limited resources and time, an increase in workload, provider beliefs, and low interest, motivation and prioritization [63-66]. At the outset, there may be large time and financial commitments as key personnel are trained in the four components and clinics experiment in how best to introduce this into the current workflow. However, implementation is facilitated by several factors such as educational opportunities for patients and providers, good collaboration and communication, good record systems, individualized treatments, and the recruitment of specialized workers [63-66].” (p 10, line 355)

  1. The discussion section should be strengthened. I suggest the authors use implementation science to discuss the facilitators and barriers to implementing PCHB, and as well as implementation strategies to enhance the facilitators and overcome the barriers. 

Thank you for this valuable suggestion. We have added a section to the discussion to address both facilitators and barriers.

“The impact of these facilitators and barriers will vary by clinic. The persons championing insomnia treatment may need to dedicate time to developing a clinic atmosphere that is receptive to changes to improve insomnia treatment. This may look like organizing training opportunities about the impacts of insomnia, best clinical practices for intervention, and referral pathways. Because a system-wide change can be costly in the front end, it could be important to get a snapshot of how many patients at a given clinic might benefit from these services to determine how large of an impact targeting insomnia might have. Increasing patient awareness about insomnia (e.g., through passive psychoeducational handouts distributed in waiting rooms) might also be a strong beginning step.

Once the clinic is on board, the next step might be to identify 1) who is involved, 2) in which component they are involved, and 3) how personnel will communicate with one another. Considerations here may include time pressure, expertise, and at what point they encounter a patient during a visit. For instance, administration of a screener measure can take place at several timepoints throughout a patient visit. It could be included in the initial appointment paperwork or handed out by front desk staff. Alternatively, it may be more feasible to have a medical assistant or even a physician administer the measure for a quick review and warm handoff to the BHC. Initial attempts should be iterative. Take note of bottlenecks, overburdened personnel, or communication lapses. Use provider and patient feedback to make changes.” (p 10, line 365)

Reviewer 2:

As this is a review on insomnia, I believe that the authors should provide at least its definition. 

We agree and appreciate you highlighting this oversight. We have now included the definition in the first paragraph of the introduction.

“The DSM-5R defines insomnia as the difficulty falling asleep, staying asleep, or waking up too early despite adequate sleep opportunity” (Page 1, line 26).

Also, it would be beneficial to provide differences in insomnia and sleep architecture across life span, and whether interventions should differ according to the patient’s different age group. 

We believe this is an excellent suggestion that enhances the quality of the manuscript. Accordingly, we have added a paragraph in the discussion section to reflect this.

“Sleep generally, and insomnia specifically, looks different across the lifespan. Older adults experience less deep sleep and more wake after sleep onset and light sleep as they age [53]. They may also report more sleep disturbances related to medication use or comorbid medical concerns [53]. Children, on the other hand, need substantially more sleep, including daytime naps [54]. Sleep disorders like insomnia are not uncommon in children. However, they may be underdiagnosed in primary care, possibly because of minimal screening and lack of caregiver awareness [55].

While behavioral interventions to treat insomnia are still recommended over the use of medications, there are different treatment modifications to consider [56]. For instance, older adults have a high risk for falls, thus the stimulus control recommendations to get out of bed if awake may not be safe. Instead, a patient might consider modifications such as countercontrol [57]. Rather than getting out of bed, patients create a clear distinction between their posture and environment when in bed to sleep versus when they are awake in bed.

When working with children, screening procedures may vary. It is recommended to have the patient and their caregiver complete a sleep diary to facilitate discussions about sleep [56]. It is also important to inquire about caregiver support in sleep-related behaviors [58, 59]. When treating insomnia in children and adolescents, behavioral interventions are effective [56, 60, 61]. Notably, sleep restriction is typically implemented only after other behavioral strategies have been exhausted [62].” (p. 10, line 333)

Reviewer 2 Report

Comments and Suggestions for Authors

Comments to the authors

This is an interesting perspective on the role and aid of BHC to implement screening and treatment for insomnia of patients who visit primary care clinics. 

Only minor point. 

As this is a review on insomnia, I believe that the authors should provide at least its definition. 

Also, it would be beneficial to provide differences in insomnia and sleep architecture across life span, and whether interventions should differ according to the patient’s different age group. 

Author Response

We sincerely appreciate the thoughtful feedback provided by the reviewers. Their insightful suggestions have greatly contributed to improving the clarity and quality of this manuscript. We are grateful for their time and effort in helping to strengthen our work. Below we address each set of comments.

Reviewer 1

  1. There are 4 components of PCBH mentioned in the paper: 1) identifying sleep issues, 2) providing psychoeducation on the possible treatments for insomnia, 3) intervening with poor sleep habits and acute insomnia early to prevent chronic insomnia, and 4) delivering appropriate evidence-based interventions for chronic insomnia. 

Is this 4-component an evidence-based approach? Or there is evidence for individual component and the authors suggested combining the 4 components to treat insomnia using PCBH? 

If the 4-component is an evidence-based approach already, please give more references for it and discuss how it has been implemented in different countries/regions. 

If it is the latter case, the authors should point out this and have a brief discussion about future research studies that should investigate the efficacy of the 4-component approach in treating insomnia. 

Thank you for your insightful feedback. The proposed components are grounded in a robust body of research, with each element supported by individual studies. We have synthesized our understanding of the benefits of PCBH and insomnia treatment to connect existing insomnia research to the PCBH framework. For example, screening is a common practice for Behavioral Health Consultants (BHCs), and in the manuscript, we aim to provide specific examples of evidence-based measures to aid in this endeavor. To make this clear, we have added an additional paragraph in the discussion section explaining the limitation and suggesting future research.

“Lastly, this review is designed to integrate the evidence for PCBH and insomnia treatment and present it in such a way that it can be implemented in the real world. We suggested the components based on the existing evidence, but future research is needed. It is important to determine if the four components are weighted equally in terms of impact and burden. Such an investigation will guide clinics to focus resources on the component with the largest benefit. Additionally, it would be important to understand how each component may interact with one another. While there is good work that supports each individual component, to our knowledge there is no work that examines these components (or similar) as a whole.” (p. 11, line 397)

  1. The authors should discuss the cost or limitations related to implementing PCBH in primary care settings, especially financial costs, such as costs related to training PCBH professionals, the feasibility of organizing such service in the primary care settings, how it could be integrated into the service provided by physicians or nurses, etc.   

We concur that this addition significantly strengthens the manuscript. Accordingly, we have incorporated a dedicated section in the discussion to address this important issue.

“Getting started with targeted insomnia assessment and treatment is a challenging undertaking for any clinic. When initiating similar programs, common barriers include limited awareness and knowledge on the part of patients and providers, limited resources and time, an increase in workload, provider beliefs, and low interest, motivation and prioritization [63-66]. At the outset, there may be large time and financial commitments as key personnel are trained in the four components and clinics experiment in how best to introduce this into the current workflow. However, implementation is facilitated by several factors such as educational opportunities for patients and providers, good collaboration and communication, good record systems, individualized treatments, and the recruitment of specialized workers [63-66].” (p 10, line 355)

  1. The discussion section should be strengthened. I suggest the authors use implementation science to discuss the facilitators and barriers to implementing PCHB, and as well as implementation strategies to enhance the facilitators and overcome the barriers. 

Thank you for this valuable suggestion. We have added a section to the discussion to address both facilitators and barriers.

“The impact of these facilitators and barriers will vary by clinic. The persons championing insomnia treatment may need to dedicate time to developing a clinic atmosphere that is receptive to changes to improve insomnia treatment. This may look like organizing training opportunities about the impacts of insomnia, best clinical practices for intervention, and referral pathways. Because a system-wide change can be costly in the front end, it could be important to get a snapshot of how many patients at a given clinic might benefit from these services to determine how large of an impact targeting insomnia might have. Increasing patient awareness about insomnia (e.g., through passive psychoeducational handouts distributed in waiting rooms) might also be a strong beginning step.

Once the clinic is on board, the next step might be to identify 1) who is involved, 2) in which component they are involved, and 3) how personnel will communicate with one another. Considerations here may include time pressure, expertise, and at what point they encounter a patient during a visit. For instance, administration of a screener measure can take place at several timepoints throughout a patient visit. It could be included in the initial appointment paperwork or handed out by front desk staff. Alternatively, it may be more feasible to have a medical assistant or even a physician administer the measure for a quick review and warm handoff to the BHC. Initial attempts should be iterative. Take note of bottlenecks, overburdened personnel, or communication lapses. Use provider and patient feedback to make changes.” (p 10, line 365)

Reviewer 2:

As this is a review on insomnia, I believe that the authors should provide at least its definition. 

We agree and appreciate you highlighting this oversight. We have now included the definition in the first paragraph of the introduction.

“The DSM-5R defines insomnia as the difficulty falling asleep, staying asleep, or waking up too early despite adequate sleep opportunity” (Page 1, line 26).

Also, it would be beneficial to provide differences in insomnia and sleep architecture across life span, and whether interventions should differ according to the patient’s different age group. 

We believe this is an excellent suggestion that enhances the quality of the manuscript. Accordingly, we have added a paragraph in the discussion section to reflect this.

“Sleep generally, and insomnia specifically, looks different across the lifespan. Older adults experience less deep sleep and more wake after sleep onset and light sleep as they age [53]. They may also report more sleep disturbances related to medication use or comorbid medical concerns [53]. Children, on the other hand, need substantially more sleep, including daytime naps [54]. Sleep disorders like insomnia are not uncommon in children. However, they may be underdiagnosed in primary care, possibly because of minimal screening and lack of caregiver awareness [55].

While behavioral interventions to treat insomnia are still recommended over the use of medications, there are different treatment modifications to consider [56]. For instance, older adults have a high risk for falls, thus the stimulus control recommendations to get out of bed if awake may not be safe. Instead, a patient might consider modifications such as countercontrol [57]. Rather than getting out of bed, patients create a clear distinction between their posture and environment when in bed to sleep versus when they are awake in bed.

When working with children, screening procedures may vary. It is recommended to have the patient and their caregiver complete a sleep diary to facilitate discussions about sleep [56]. It is also important to inquire about caregiver support in sleep-related behaviors [58, 59]. When treating insomnia in children and adolescents, behavioral interventions are effective [56, 60, 61]. Notably, sleep restriction is typically implemented only after other behavioral strategies have been exhausted [62].” (p. 10, line 333)

Round 2

Reviewer 1 Report

Comments and Suggestions for Authors

Thank you for addressing the comments.